# Breaking an Image Encryption Algorithm Based on DNA Encoding and Spatiotemporal Chaos

**DOI:** 10.3390/e21030246

**Published:** 2019-03-05

**Authors:** Heping Wen, Simin Yu, Jinhu Lü

**Affiliations:** 1School of Automation, Guangdong University of Technology, Guangzhou 510006, China; 2School of Automation Science and Electrical Engineering, State Key Laboratory of Software Development Environment, and Beijing Advanced Innovation Center for Big Data and Brain Computing, Beihang University, Beijing 100191, China

**Keywords:** image encryption, DNA encoding, chaotic cryptography, cryptanalysis, image privacy

## Abstract

Recently, an image encryption algorithm based on DNA encoding and spatiotemporal chaos (IEA-DESC) was proposed. In IEA-DESC, pixel diffusion, DNA encoding, DNA-base permutation and DNA decoding are performed successively to generate cipher-images from the plain-images. Some security analyses and simulation results are given to prove that it can withstand various common attacks. However, in this paper, it is found that IEA-DESC has some inherent security defects as follows: (1) the pixel diffusion is invalid for attackers from the perspective of cryptanalysis; (2) the combination of DNA encoding and DNA decoding is equivalent to bitwise complement; (3) the DNA-base permutation is actually a fixed position shuffling operation for quaternary elements, which has been proved to be insecure. In summary, IEA-DESC is essentially a combination of a fixed DNA-base position permutation and bitwise complement. Therefore, IEA-DESC can be equivalently represented as simplified form, and its security solely depends on the equivalent secret key. So the equivalent secret key of IEA-DESC can be recovered using chosen-plaintext attack and chosen-ciphertext attack, respectively. Theoretical analysis and experimental results show that the two attack methods are both effective and efficient.

## 1. Introduction

With the rapid development of information technologies such as mobile Internet, cloud computing, social networking, and Big Data, the security of multimedia data such as image and video has attracted more and more attention [1,2,3]. Image is an important part of multimedia data, and its encryption protection techniques are particularly interesting [4,5]. In the past three decades, many novel image encryption schemes based on various methodologies were proposed, such as chaos theory [6], DNA computing [7], cellular automaton [8,9], and quantum information [9,10]. Among them, chaos is the most popular one because it has the unique characteristics of sensitivity to initial values and parameters, ergodicity, and deterministic inherent randomness [11,12,13,14,15,16], which correspond to the confusion and diffusion properties of encryption [17]. Moreover, DNA computing has the characteristics of high parallelism, large storage capacity, and low energy consumption [7]. Hence, researches on image encryption schemes combined with chaos theory and DNA computing have become a hot topic in recent years [18,19,20]. Nevertheless, many encryption schemes are actually insecure as a result of their various security defects [21,22]. Therefore, performing cryptanalysis on these existing encryption algorithms is indispensable [21,23,24].

In recent years, with the security analysis and breaking of some existing chaotic image algorithms combining DNA computing and chaos theory [25,26,27,28,29,30], research interest in cryptanalysis has become increasingly stimulated [31,32,33,34]. In 2010, Zhang et al. [25] created an image encryption method using DNA addition combined with chaotic maps. However, in 2014, Hermassi et al. [26] pointed out that the algorithm in Reference [25] was irreversible and was vulnerable to the chosen-plaintext attack and the known-plaintext attack. In 2015, Zhen et al. [27] proposed an image encryption scheme combining DNA coding and entropy. Nonetheless, in 2016, Su et al. [28] pointed out that the algorithm of Reference [27] was insecure and could be broken using the chosen-plaintext attack. In 2016, Jain et al. [29] proposed a robust DNA chaotic image encryption scheme based on Reference [25] and Reference [26]. Whereas, in 2017, Dou et al. [30] used the chosen-plaintext attack method to break the algorithm proposed in Reference [29]. In addition, Özkaynak et al. [33] and Zhang et al. [34] further concluded that an encryption algorithm may lead to the existence of an equivalent secret key if only a single DNA encoding and operation rule is employed.

Generally speaking, cryptanalysis becomes more difficult as the level of encryption design increases [35]. However, there are still some existing algorithms that can be broken owing to their inherent defects [36]. Moreover, since each encryption algorithm has natural features, the corresponding attack method may also be different. Therefore, it makes sense, even if a similar attack method is used, to reveal the intrinsic characteristics of the different encryption algorithms.

In 2015, an image encryption algorithm based on DNA encoding and spatiotemporal chaos (IEA-DESC) was proposed [37]. In IEA-DESC, pixel diffusion, DNA encoding, DNA-base permutation, and DNA decoding are adopted successively to obtain cipher-images from plain-images. Some security analyses and simulation results are given to prove that it can withstand various common attacks. Despite this, according to the basic criteria of cryptanalysis, some findings in IEA-DESC can be given as follows:(1)Its pixel diffusion is invalid for attackers.

In IEA-DESC, there is no external secret key during the pixel diffusion phase. According to the cryptographic principle proposed by Kerckhoffs [38], the algorithm is public for attackers. Therefore, its pixel diffusion is essentially useless.
(2)The combination of DNA encoding and DNA decoding can be equivalently simplified.

Although the DNA encoding rule is related to the plain-image, there is a certain relationship between its decoding rule and its encoding rule. This leads to the fact that for any binary bit, the output is the complement of the input after DNA encoding and DNA decoding. Hence, DNA encoding and DNA decoding are a complementary process on the whole.
(3)The sequences for DNA-base permutation are fixed for different plain-images.

During IEA-DESC’s DNA-base permutation, the chaos-based sequences for encryption are neither associated with plain-image nor cipher-image. Thus, on the basis of the basic rules of cryptanalysis, under the condition of a given secret key, the encryption sequences are fixed for different plain-images. Once the attackers obtain these sequences, i.e., an equivalent secret key, the DNA-base permutation is deciphered.

On the basis of the above properties, IEA-DESC’s pixel diffusion is invalid, and therefore, its security depends only on the DNA domain encryption. Unfortunately, an equivalent secret key exists in the overall DNA domain encryption phase. More specifically, the DNA-based encryption algorithm is essentially a permutation-only process of a quaternary element. Yet, permutation-only encryption algorithms have been analyzed to be insecure [39,40]. Therefore, in this paper, two attack methods for breaking IEA-DESC using the chosen-plaintext attack and chosen-ciphertext attack are proposed, respectively.

The rest of the paper is organized as follows. Section 2 concisely describes IEA-DESC. Section 3 proposes two different attack methods on IEA-DESC. Section 4 presents the experimental simulation results. Section 5 gives some improvement suggestions for the security of chaos-based encryption algorithms. The last section concludes the paper.

## 2. The Encryption Algorithm under Study

In this section, the DNA coding rules and spatiotemporal chaos used in Reference [37] are introduced, and then the specific steps of IEA-DESC are detailed.

### 2.1. DNA Coding Rules

A DNA sequence includes four kinds of nucleic acid bases: A, T, C, and G. With respect to these four bases, the total number of coding combinations is 4!=24. However, there are only eight kinds of coding combinations because these four bases satisfy the principle of complementary base pairs. More precisely, A and T are complementary to each other, as are C and G. Table 1 shows the eight DNA coding rules.

### 2.2. Spatiotemporal Chaos

Two discrete chaotic maps are used in IEA-DESC [37], one is the logistic map and the other is a spatiotemporal chaos map based on the so-called new chaotic algorithm (NCA) given in Reference [41]. The iterative equation of the Logistic map is represented as
(1)$$x_{n+1}=\mu x_{n}\left(1-x_{n}\right),$$
where the state variable x∈(0,1) and the control parameter μ∈(3.57,4). The structure of the functional graph of the Logistic map in a digital computer is quantitatively analyzed in Reference [16].

The spatiotemporal chaos is a dynamic system using discrete time and space, in which the coupled map lattice (CML) is its most common model. The iterative equation of NCA-based CML is modeled by
(2)$$\left\{\begin{array}{c} x_{n+1}\left(i\right)=\left(1-\varepsilon \right)f\left(x_{n}\left(i\right)\right)+\varepsilon \left\{f\left[x_{n}\left(i-1\right)\right]+f\left[x_{n}\left(i+1\right)\right]\right\}/2,\\ f\left(x\right)=\left(1-\beta^{-4}\right)\cdot \mathrm{ctg}\left(\alpha /\left(1+\beta \right)\right)\cdot {\left(1+1/\beta \right)}^{\beta }\cdot \mathrm{tg}\left(\alpha x_{n}\right)\cdot {\left(1-x\right)}^{\beta }, \end{array}\right.$$
where the spatial lattice index i=1,2,⋯,L, the time grid index n=1,2,⋯, the coupling strength ε∈(0,1), the state variable $x_{n}\left(i\right)\in \left(0,1\right)$, and the periodic boundary condition is $x_{n}\left(0\right)=x_{n}\left(L\right)$. The second equation of Equation (Equation 2) is the so-called NCA, which is actually an improved logistic map. Given the parameters α=1.57,β=3.5,ε=0.3, and L=1024, the system is chaotic, and its attractor is shown in Figure 1.

### 2.3. Description of IEA-DESC

#### 2.3.1. Secret Key

The secret key of IEA-DESC consists of $x_{0}$, $\mu ,K_{0},N_{0},\alpha ,\beta ,\varepsilon$, and *L*, where $x_{0},\mu ,K_{0}$, and $N_{0}$ are the parameters of the logistic map, α,β,ε, and *L* are the parameters of NCA-based CML, and $N_{0}$ is the length of discarded sequence for eliminating harmful transient effects.

#### 2.3.2. Encryption Process

The encryption objects of IEA-DESC are 8-bit grayscale images of size H×W (height × width). For convenience, the symbolic representation is different without changing the original algorithm. A block diagram of IEA-DESC is shown in Figure 2, where P, $P^{\prime }$, and C are the plain-image, the diffused image, and the cipher-image, respectively. As can be seen from Figure 2, the encryption process of IEA-DESC includes four phases: pixel diffusion, DNA encoding, DNA-base permutation, and DNA decoding.

The specific descriptions of IEA-DESC are given as follows:**Phase 1.** Pixel Diffusion:By converting the plain-image P into the corresponding sequence $\left\{p_{1},p_{2},\ldots ,p_{H\times W}\right\}$ in raster scanning order, the pixel diffusion equation is defined as
(3)$$\left\{\begin{array}{c} p_{1}^{\prime }=p_{1}⊕p_{H\times W},\\ p_{i+1}^{\prime }=p_{i}⊕p_{i}^{\prime }, \end{array}\right.$$
where i=1,2,…,H×W−1, and ⊕ represents the bitwise XOR operation. Thus, the diffused image $P^{\prime }$ of size H×W is obtained from the diffused sequence $\left\{p_{1}^{\prime },p_{2}^{\prime },\ldots ,p_{H\times W}^{\prime }\right\}$. **Phase 2.** DNA Encoding:Calculating the sum of the plain-image pixels, the $K_{0}$-th iteration value $x_{K_{0}}$ is obtained by Equation (Equation 1) under the initial value $x_{0}$ and the control parameter μ. The DNA encoding rule $r_{E}$, as in Table 1, is further determined by
(4)$$r_{E}=\left\lfloor x_{K_{0}}\times 8\right\rfloor +1,$$
where $r_{E}\in \left[1,8\right]$, and ⌊a⌋ rounds the element *a* to the nearest integer toward minus infinity. Then, by the $r_{E}$-th encoding rule in Table 1, the diffused image $P^{\prime }$ of size H×W is firstly converted into the corresponding binary matrix of size H×8W, and then encoded as the DNA matrix $D_{1}$ of size H×4W.**Phase 3.** DNA-Base Permutation:First, by iterating Equation (Equation 1) $N_{0}+4W$ times and then discarding the front $N_{0}$ elements under the initial value $x_{0}$ and the control parameter μ, a sequence $\left\{a_{1},a_{2},\ldots ,a_{4W}\right\}$ of length 4W is obtained. Here, the sequence $\left\{a_{1},a_{2},\ldots ,a_{4W}\right\}$ is taken as an initial value of the spatiotemporal chaos called NCA-based CML. Thus, by iterating Equation (Equation 2) *H* times under the parameters α,β,ε, and *L*, a real matrix X of size H×4W is achieved.Then, by sorting each row’s elements of X in ascending order, the corresponding *H* row position index sequences are obtained as ${\mathrm{RI}}_{i}$. Using ${\mathrm{RI}}_{i}$ to perform permutation for each row on the DNA matrix $D_{1}$, the corresponding row permuted DNA matrix $D_{1}^{\prime }$ is obtained, given by
(5)$${\left[D_{1}^{\prime }\right]}_{i,k}={\left[D_{1}\right]}_{i,{\mathrm{RI}}_{i}\left(k\right)},$$
where i=1,2,…,H, k=1,2,…,4W, ${\mathrm{RI}}_{i}\left(k\right)$ indicates a position index of the *k*-th element in the *i*-th row, and ${\mathrm{RI}}_{i}\left(k\right)\in \left\{1,2,\ldots ,4W\right\}$. Similarly, by sorting each column’s elements of X in ascending order, the corresponding 4W column position index sequences are obtained as ${\mathrm{CI}}_{j}$. Using ${\mathrm{CI}}_{j}$ to perform permutation for each column on the DNA matrix $D_{1}^{\prime }$, the corresponding column permuted DNA matrix $D_{2}$ is obtained, represented as
(6)$${\left[D_{2}\right]}_{k,j}={\left[D_{1}^{\prime }\right]}_{{\mathrm{CI}}_{j}\left(k\right),j},$$
where j=1,2,…,4W, k=1,2,…,H, ${\mathrm{CI}}_{j}\left(k\right)$ indicates a position index of the *k*-th element in the *j*-th column, and ${\mathrm{CI}}_{j}\left(k\right)\in \left\{1,2,\ldots ,H\right\}$.**Phase 4.** DNA Decoding:Corresponding to Equation (Equation 4), the DNA decoding rule $r_{D}$ is determined as
(7)$$r_{D}=9-r_{E},$$
where $r_{D}\in \left[1,8\right]$. Thus, by the $r_{D}$-th decoding rule in Table 1, the DNA matrix $D_{2}$ of size H×4W is firstly decoded as the corresponding binary matrix of size H×8W, and then converted into the cipher-image C of size H×W.

#### 2.3.3. Decryption Process

Decryption is the inverse of encryption. First, the cipher-image C is converted into the DNA matrix $D_{2}$ by the $r_{D}$-th encoding rule. Then, the DNA matrix $D_{1}$ is exacted from the DNA matrix $D_{2}$ after the anti-permutation. Next, the DNA matrix $D_{1}$ is decoded as the diffused image $P^{\prime }$ with the $r_{E}$-th decoding rule. Finally, the plain-image P is recovered by anti-diffusion decryption from Equation (Equation 3).

## 3. Cryptanalysis of IEA-DESC

### 3.1. Preliminary Analysis of IEA-DESC

According to modern cryptography principles, encryption algorithms are public and only the secret keys are unknown to attackers [42,43]. More precisely, the security of an algorithm solely depends on its secret key. Four common attack methods for cryptanalysis are shown in Table 2. A secure cryptosystem should be able to resist all types of attacks in Table 2. If a cryptosystem cannot resist anyone of these attacks, one can conclude that the cryptosystem is insecure.

By observing Figure 2, one can divide the encryption process of IEA-DESC into two parts, one is pixel diffusion, and the other is DNA domain encryption. For the pixel diffusion part, there is no secret key involved. Since the algorithm is open from the perspective of cryptanalysis, the diffusion phase of IEA-DESC is essentially invalid for the attacker.

For this reason, only the DNA domain encryption part is worthy of further discussion here. Following Equations (Equation 4) and (Equation 7), one knows that there is a definite one-to-one correspondence between DNA encoding rule $r_{E}$ and DNA decoding rule $r_{D}$. Hence, one can list all possible pairs of DNA codec rules, given as
(8)$$\left(r_{E},r_{D}\right)\in \left\{(1,8),(2,7),(3,6),(4,5),(5,4),(6,3),(7,2),(8,1)\right\}.$$

Accordingly, given the eight pairs of DNA codec rules, within the binary bits before and after DNA coding, there appears a certain regularity [34]. Table 3 shows any 2-bit input and its 2-bit output with the eight pairs of DNA codec rules. As can be seen from Table 3, given any 2-bit input, no matter which DNA encoding rule is taken, the corresponding 2-bit output is the same because $r_{E}+r_{D}=9$ holds. Put explicitly, the 2-bit input and the corresponding 2-bit output are complementary.

Furthermore, one can see that the chaos-based sequences for DNA-base permutation are fixed for different plain-images under the premise of a given secret key. Indeed, it means that an equivalent secret key exists in IEA-DESC.

On this basis, it is found that IEA-DESC is essentially a combination process of a fixed DNA-base position permutation and bitwise complement. Therefore, a simplified block diagram of IEA-DESC can be illustrated, as is shown in Figure 3, where $EK_{P}$ is the equivalent secret key. Once $EK_{P}$ is obtained, IEA-DESC will be broken. Note that the eight different pairs of DNA encoding and decoding, as in Table 3, are equivalent. For simplicity, one sets $r_{E}=1$ and $r_{D}=8$, i.e., the first DNA encoding rule and the eighth DNA decoding rule are adopted below.

### 3.2. Analysis of DNA-Base Permutation

To obtain the equivalent secret key $EK_{P}$, performing analysis on the DNA-base permutation is significant. Since DNA only has four different bases, the essence of DNA-base permutation is a position shuffling procedure for a quaternary matrix of size H×4W.

Supposing that one has an input matrix PV of size H×4W which satisfies that each element is unequal, and its corresponding one-dimensional sequence in the raster scanning order is $\left\{0,1,2,\ldots ,4HW-1\right\}$. Letting $Z_{m}$ denote a set $\left\{0,1,\cdots ,m-1\right\}$, one has $\mathrm{PV}\in Z_{4HW}$. Obviously, after a position permutation, the output matrix CV corresponding to the input matrix PV also has the feature that all elements are not equal to each other. According to the assumption of the chosen-plaintext attack in Table 2, one can know both PV and CV. Thus, the equivalent secret key can be determined by comparing the elements before and after the permutation.

However, since each DNA-base only takes four values, A, G, C, and T, such an input matrix PV does not exist. To cope with the problem, an appropriate transformation is inevitable. Therefore, the specific analysis steps to determine the equivalent secret key $EK_{P}$, as in Figure 3, are detailed as follows:**Step** **1.**Decompose the virtual matrix PV of size H×4W into some quaternary matrices of the same size.

The virtual matrix PV is firstly decomposed into the $N_{C}$ corresponding quaternary matrices ${\mathrm{PQ}}_{n}\left(n=1,2,\ldots ,N_{C}\right)$, defined by
(9)$$\mathrm{PV}=\sum_{n=1}^{N_{C}}4^{n-1}{\mathrm{PQ}}_{n}={\mathrm{PQ}}_{1}+4{\mathrm{PQ}}_{2}+4^{2}{\mathrm{PQ}}_{3}+\ldots +4^{N_{C}-1}{\mathrm{PQ}}_{N_{C}},$$
where $\mathrm{PV}\in Z_{4HW}$, ${\mathrm{PQ}}_{n}\in Z_{4}$, and $N_{C}$ is the minimum amount required to ensure this decomposition method. Referring to Reference [39], generally one has
(10)$$\begin{array}{cc} N_{C} & =\left\lceil \log_{4}\left(H\times 4W\right)\right\rceil \\ & =1+\left\lceil \log_{4}\left(HW\right)\right\rceil , \end{array}$$
where $\left\lceil a\right\rceil$ rounds the element *a* to the nearest integer toward positive infinity.
**Step** **2.**Transform these quaternary matrices into the 8-bit images of size H×W, respectively.

The quaternary matrices ${\mathrm{PQ}}_{n}\left(n=1,2,\ldots ,N_{C}\right)$ are transformed into the corresponding decimal matrices using the method whereby every four quaternary elements are combined into a decimal one in order from low to high. For instance, given four quaternary elements are 0, 1, 2, and 3, one gets the corresponding decimal result as 228 because of its combination procedure: $0\times 1+1\times 4+2\times 4^{2}+3\times 4^{3}$. In fact, these decimal matrices are the resulting 8-bit images ${\mathrm{PI}}_{n}\left(n=1,2,\ldots ,N_{C}\right)$ of size H×W.
**Step** **3.**Temporarily use the encryption machine to obtain the $N_{C}$ corresponding cipher-images.

Following Figure 3, the diffused image $P^{\prime }$ is deemed as an input plain-image. As for the chosen-plaintext attack in Table 2, the input plain-images can be arbitrarily chosen, and the encryption machine can be temporarily used. Therefore, one gets the $N_{C}$ cipher-images ${\mathrm{CI}}_{n}\left(n=1,2,\ldots ,N_{C}\right)$ corresponding to the input plain-images ${\mathrm{PI}}_{n}\left(n=1,2,\ldots ,N_{C}\right)$, respectively, after the encryption. Obviously, one has ${\mathrm{PI}}_{n}\in Z_{256}$ and ${\mathrm{CI}}_{n}\in Z_{256}$.

As shown in Figure 3, it takes three phases from ${\mathrm{PI}}_{n}$ to ${\mathrm{CI}}_{n}$: DNA encoding, DNA-base permutation, and DNA decoding. First, the $N_{C}$ plain-images ${\mathrm{PI}}_{n}$ are encoded as the corresponding DNA matrices with the first DNA encoding rule of Table 1. Then, the permuted DNA matrices are further obtained after a DNA-base permutation. Finally, the $N_{C}$ corresponding cipher-images ${\mathrm{CI}}_{n}$ are decoded from the permuted DNA matrices with the eighth DNA decoding rule.
**Step** **4.**Convert these 8-bit cipher-images into the quaternary matrices of size H×4W, respectively.

Note that the eighth DNA decoding rule corresponds to the first DNA encoding rule, and the combination process of DNA encoding and DNA decoding is bitwise complement analyzed as in Table 3. Therefore, the complement operation cannot be ignored.

First, similar to the method in Step 2, the $N_{C}$ 8-bit cipher-images ${\mathrm{CI}}_{n}\left(n=1,2,\ldots ,N_{C}\right)$ of size H×W are converted to the corresponding quaternary matrices ${\mathrm{CQ}}_{n}\left(n=1,2,\ldots ,N_{C}\right)$ of size H×4W. Then, the corresponding complementary quaternary matrices ${\bar{\mathrm{CQ}}}_{n}\left(n=1,2,\ldots ,N_{C}\right)$ can be obtained from these quaternary matrices ${\mathrm{CQ}}_{n}\left(n=1,2,\ldots ,N_{C}\right)$, respectively. Here, the quaternary complement operation is defined as being subtracted by 3. Specifically, the complements of 0, 1, 2, and 3 are 3, 2, 1, and 0, respectively.
**Step** **5.**Compose the $N_{C}$ complementary quaternary matrices into a virtual matrix of size H×4W.

Corresponding to Equation (Equation 9), the virtual matrix CV is composed from the $N_{C}$ complementary quaternary matrices ${\bar{\mathrm{CQ}}}_{n}\left(n=1,2,\ldots ,N_{C}\right)$, given as
(11)$$\mathrm{CV}=\sum_{n=1}^{N_{C}}4^{n-1}{\bar{\mathrm{CQ}}}_{n}={\bar{\mathrm{CQ}}}_{1}+4{\bar{\mathrm{CQ}}}_{2}+4^{2}{\bar{\mathrm{CQ}}}_{3}+\ldots +4^{N_{C}-1}{\bar{\mathrm{CQ}}}_{N_{C}},$$
where $\mathrm{CV}\in Z_{4HW}$, and ${\bar{\mathrm{CQ}}}_{n}\in Z_{4}$.
**Step** **6.**Obtain the equivalent secret key $EK_{P}$.

Finally, $EK_{P}$ is obtained by comparing all the different elements of PV and CV.

To better illustrate this analysis process, a simple example is taken. Let a input virtual matrix PV of size 4×4 be
$$\mathrm{PV}=\left[\begin{array}{cccc} 0 & 1 & 2 & 3\\ 4 & 5 & 6 & 7\\ 8 & 9 & 10 & 11\\ 12 & 13 & 14 & 15 \end{array}\right].$$

First, following Steps 1 and 2, one gets two quaternary matrices ${\mathrm{PQ}}_{1}$ and ${\mathrm{PQ}}_{2}$, the two 8-bit images ${\mathrm{PI}}_{1}$ and ${\mathrm{PI}}_{2}$, and their corresponding DNA matrices as below:$${\mathrm{PQ}}_{1}=\left[\begin{array}{cccc} 0 & 1 & 2 & 3\\ 0 & 1 & 2 & 3\\ 0 & 1 & 2 & 3\\ 0 & 1 & 2 & 3 \end{array}\right]\to {\mathrm{PI}}_{1}=\left[\begin{array}{c} 228\\ 228\\ 228\\ 228 \end{array}\right]\overset{r_{E}=1}{\to }\left[\begin{array}{cccc} A & G & C & T\\ A & G & C & T\\ A & G & C & T\\ A & G & C & T \end{array}\right],$$
$${\mathrm{PQ}}_{2}=\left[\begin{array}{cccc} 0 & 0 & 0 & 0\\ 1 & 1 & 1 & 1\\ 2 & 2 & 2 & 2\\ 3 & 3 & 3 & 3 \end{array}\right]\to {\mathrm{PI}}_{2}=\left[\begin{array}{c} 0\\ 85\\ 170\\ 255 \end{array}\right]\overset{r_{E}=1}{\to }\left[\begin{array}{cccc} A & A & A & A\\ G & G & G & G\\ C & C & C & C\\ T & T & T & T \end{array}\right].$$

Then, as in Steps 3 and 4, supposing that the procedure of DNA-base permutation is given in Figure 4, one obtains the two cipher-images ${\mathrm{CI}}_{1}$ and ${\mathrm{CI}}_{2}$, their corresponding DNA matrices, and the two quaternary ones ${\mathrm{CQ}}_{1}$ and ${\mathrm{CQ}}_{2}$ via
$${\mathrm{CQ}}_{1}=\left[\begin{array}{cccc} 2 & 1 & 2 & 0\\ 1 & 3 & 2 & 3\\ 0 & 1 & 2 & 3\\ 0 & 1 & 3 & 0 \end{array}\right]\leftarrow {\mathrm{CI}}_{1}=\left[\begin{array}{c} 38\\ 237\\ 228\\ 52 \end{array}\right]\overset{r_{D}=8}{\to }\left[\begin{array}{cccc} G & C & G & T\\ C & A & G & A\\ T & C & G & A\\ T & C & A & T \end{array}\right],$$
$${\mathrm{CQ}}_{2}=\left[\begin{array}{cccc} 0 & 0 & 1 & 0\\ 1 & 2 & 2 & 3\\ 1 & 2 & 3 & 0\\ 2 & 3 & 1 & 3 \end{array}\right]\leftarrow {\mathrm{CI}}_{2}=\left[\begin{array}{c} 16\\ 233\\ 57\\ 222 \end{array}\right]\overset{r_{D}=8}{\to }\left[\begin{array}{cccc} T & T & C & T\\ C & G & G & A\\ C & G & A & T\\ G & A & C & A \end{array}\right].$$

Correspondingly, one gets the two complementary quaternary matrices ${\bar{\mathrm{CQ}}}_{1}$ and ${\bar{\mathrm{CQ}}}_{2}$ as
$${\bar{\mathrm{CQ}}}_{1}=\left[\begin{array}{cccc} 1 & 2 & 1 & 3\\ 2 & 0 & 1 & 0\\ 3 & 2 & 1 & 0\\ 3 & 2 & 0 & 3 \end{array}\right],{\bar{\mathrm{CQ}}}_{2}=\left[\begin{array}{cccc} 3 & 3 & 2 & 3\\ 2 & 1 & 1 & 0\\ 2 & 1 & 0 & 3\\ 1 & 0 & 2 & 0 \end{array}\right].$$

Next, as in Step 5, one obtains the corresponding output virtual matrix as
$$\mathrm{CV}=\left[\begin{array}{cccc} 13 & 14 & 9 & 15\\ 10 & 4 & 5 & 0\\ 11 & 6 & 1 & 12\\ 7 & 2 & 8 & 3 \end{array}\right].$$

Finally, the equivalent secret key $EK_{P}$ is achieved with Step 6.

On the basis of the above discussion, one can conclude that the encryption algorithm given in Figure 3 can be broken just with the equivalent secret key $EK_{P}$ without knowing any secret key parameter.

### 3.3. Breaking IEA-DESC Using the Chosen-Plaintext Attack

Following Section 3.2, the diffused image $P^{\prime }$ is considered as the input of the cryptosystem. However, as shown in Figure 2, the actual input of IEA-DESC is the plain-image P rather than the diffused image $P^{\prime }$. To accommodate to this change, the input chosen plain-image should be adjusted accordingly.

According to the analysis in Section 3.1, the pixel diffusion part of IEA-DESC is actually useless for attackers. Under the premise that the algorithm is known, there is a certain one-to-one correspondence between the diffused image and the plain-image. Therefore, for 8-bit grayscale images of size H×W, the specific analysis steps for the chosen-plaintext attack are given as follows:**Step** **1.**Choose some special plain-images.

The $N_{C}$ 8-bit images ${\mathrm{PI}}_{n}\left(n=1,2,\ldots ,N_{C}\right)$ constructed in Section 3.2 are presented as the diffused images $P_{n}^{\prime }\left(n=1,2,\ldots ,N_{C}\right)$, respectively, and then their one-to-one corresponding plain-images $P_{n}\left(n=1,2,\ldots ,N_{C}\right)$ are obtained using anti-diffusion decryption, which is defined from Equation (Equation 3) as
(12)$$\left\{\begin{array}{c} p_{i}=p_{i+1}^{\prime }⊕p_{i}^{\prime },\\ p_{H\times W}=p_{1}⊕p_{1}^{\prime }, \end{array}\right.$$
where i=1,2,…,H×W−1, $\left\{p_{1},p_{2},\ldots ,p_{H\times W}\right\}$ and $\left\{p_{1}^{\prime },p_{2}^{\prime },\ldots ,p_{H\times W}^{\prime }\right\}$ are the sequences transformed by the plain-image P and the diffused image $P^{\prime }$ in the raster scanning order, respectively.
**Step** **2.**Temporarily use the encryption machine to get the corresponding cipher-images.

On the basis of the condition of the chosen-plaintext attack, the corresponding $N_{C}$ cipher-images $C_{n}\left(n=1,2,\ldots ,N_{C}\right)$ are obtained from the $N_{C}$ plain-images $P_{n}\left(n=1,2,\ldots ,N_{C}\right)$ by temporarily using the encryption machine.
**Step** **3.**Achieve the equivalent DNA-base permutation secret key.

By substituting ${\mathrm{PI}}_{n}\left(n=1,2,\ldots ,N_{C}\right)$ and ${\mathrm{CI}}_{n}\left(n=1,2,\ldots ,N_{C}\right)$ in Section 3.2 with the diffused images $P_{n}^{\prime }\left(n=1,2,\ldots ,N_{C}\right)$ and the cipher-images $C_{n}\left(n=1,2,\ldots ,N_{C}\right)$, respectively, one gets the equivalent DNA-base permutation secret key $EK_{P}$ with the same method as in Section 3.2.
**Step** **4.**Recover the images with the equivalent secret key.

First, using the equivalent secret key $EK_{P}$, the corresponding diffused image can be obtained from a cipher-image. Then, the recovered plain-image is obtained from the diffused images with Equation (Equation 12).

Therefore, the chosen-plaintext attack is effective to break IEA-DESC, and its data complexity is $O(N_{C})$ = $O(1+\left\lceil \log_{4}\left(HW\right)\right\rceil )$.

### 3.4. Breaking IEA-DESC Using the Chosen-Ciphertext Attack

Since the encryption structure of Figure 3 is symmetrical, the chosen-ciphertext attack is also available. The specific analysis steps based on the chosen-ciphertext attack are detailed below:**Step** **1.**Choose some specific cipher-images and temporarily use the decryption machine to get the corresponding plain-images.

Here, the $N_{C}$ images ${\left\{{\mathrm{PI}}_{n}\right\}}_{n=1}^{N_{C}}$ in Secction Section 3.2 are served as the chosen cipher-images ${\left\{C_{n}\right\}}_{n=1}^{N_{C}}$ respectively, and then temporarily use the decryption machine to get the corresponding plain-images ${\left\{P_{n}\right\}}_{n=1}^{N_{C}}$.
**Step** **2.**Get the corresponding diffused images.

The one-to-one corresponding $N_{C}$ diffused images $P_{n}^{\prime }$$\left(n=1,2,\ldots ,N_{C}\right)$ are obtained from these plain-images $P_{n}\left(n=1,2,\ldots ,N_{C}\right)$ using Equation (Equation 3).
**Step** **3.**Achieve the equivalent secret key.

The equivalent DNA-base permutation secret key is achieved by using the same method as Step 3 in Section 3.3.
**Step** **4.**Recover images with the equivalent secret key:

This step is also the same as Step 4 in Section 3.3, so it is omitted.

Therefore, the chosen-ciphertext attack is also valid for breaking IEA-DESC, and its data complexity is also $O(1+\left\lceil \log_{4}\left(HW\right)\right\rceil )$.

## 4. The Experiments for Breaking IEA-DESC

To verify the feasibility of the two proposed attack methods, some experimental simulations were performed based on a personal computer with Matlab R2016a. Similar to those in Reference [37], our experimental images are 8-bit grayscale images “Lenna” and “Peppers” of size 256×256.

### 4.1. Breaking IEA-DESC by Chosen-Plaintext Attack

The experiment for breaking IEA-DESC was firstly carried out by the chosen-plaintext attack method proposed in Section 3.3. Given H=256 and W=256, one gets $N_{C}$ = $1+\left\lceil \log_{4}\left(H\times W\right)\right\rceil$ = 9 from Equation (Equation 10). Correspondingly, the nine 8-bit images ${\mathrm{PI}}_{n}$(n=1,2,…,9) constructed using the method in Section 3.2 are shown in Figure 5a–r.

First, following Step 1 in Section 3.3, the nine special images shown in Figure 5 are selected as the diffused images $P_{n}^{\prime }\left(n=1,2,\ldots ,9\right)$, respectively, and then their corresponding plain-images $P_{n}\left(n=1,2,\ldots ,9\right)$ are obtained, as shown in Figure 6a–r. Then, according to Step 2 in Section 3.3, the nine corresponding cipher-images $C_{n}\left(n=1,2,\ldots ,9\right)$ and their histograms are obtained as shown in Figure 7a–r. Next, using the method in Step 3 in Section 3.3, the equivalent secret key $EK_{P}$ is obtained using the nine chosen diffused images shown in Figure 5a–r and the nine corresponding cipher-images shown in Figure 7a–r. Finally, the images are recovered using the equivalent secret key $EK_{P}$. The attacking results on IEA-DESC with the 8-bit images “Lenna” and “Peppers” are shown in Figure 8a–d and Figure 9a–d, respectively.

### 4.2. Breaking IEA-DESC Using the Chosen-Ciphertext Attack

Accordingly, the experiment for breaking IEA-DESC is accomplished using the chosen-ciphertext attack method proposed in Section 3.4.

First, following Step 1 in Section 3.4, the nine special images shown in Figure 5 are used as the cipher-images $C_{n}\left(n=1,2,\ldots ,9\right)$ respectively, and then their corresponding plain-images $P_{n}\left(n=1,2,\ldots ,9\right)$ are obtained, as shown in Figure 10a–r. Then, according to Step 2 in Section 3.4, the nine corresponding diffused images $P_{n}^{\prime }\left(n=1,2,\ldots ,9\right)$ and their histograms are obtained as shown in Figure 11a–r, respectively. Next, using the method in Step 3 in Section 3.4, the equivalent secret key $EK_{P}$ is obtained using the nine chosen cipher-images shown in Figure 5a–r and the nine corresponding diffused images shown in Figure 11a–r. Finally, the images are recovered using the equivalent secret key $EK_{P}$. The attacking results on IEA-DESC with the 8-bit images “Lenna” and “Peppers” are also shown in Figure 8a–d and Figure 9a–d, respectively.

### 4.3. Attack Complexity

In terms of attack complexity, the running times of the chosen-plaintext attack method and the chosen-ciphertext attack method are about 2.1165 s and 2.0785 s, respectively. Moreover, given 8-bit images of size 256×256, the data complexity of the two attack methods required for breaking IEA-DESC are both O(9). Therefore, the experimental results verify that the two attack methods are both effective and efficient.

## 5. Suggestions for Improvement

On the basis of the analysis above, IDE-DESC can neither resist against chosen-plaintext attacks nor chosen-ciphertext attacks because of its inherent security defects. In fact, some other chaos-based ciphers also have similar vulnerabilities as mentioned in Reference [36]. To deal with these problems, some suggestions for improvement to enhance the security are given below:(1)Checking the validity of each encryption component is significant.

The diffusion part of IEA-DESC is invalid for the attacker because it does not involve any secret key parameter. In fact, it does not contribute to security, but increases the computational complexity of the algorithm. Therefore, the designed algorithms should be scrutinized from the perspective of cryptanalysis to ensure the validity of each encryption component.
(2)Exploiting some novel permutation mechanisms to enhance the security.

Like other permutation-only encryption algorithms, DNA-base permutation only changes the position but does not change the value of each element. The only difference is that the element is quaternary. For permutation-only algorithms, many studies have proved that they are insecure [39,40,44]. To fulfil this demand, exploiting some novel permutation mechanisms is worthwhile.
(3)Avoiding the existence of an equivalent secret key in the algorithm.

The encryption process of the algorithm should be associated with the characteristics of the plain-image or cipher-image [2]. Otherwise, the encryption process for different input images is completely identical, which may lead to the existence of an equivalent secret key. Once the equivalent secret key is obtained by an attacker, the encryption algorithm is broken [36].
(4)Appropriately increasing the number of encryption rounds.

In a single-round encryption algorithm, the confusion and diffusion characteristics maybe insufficient [42]. Increasing the number of encryption rounds can effectively improve this problem. Of course, it also requires higher computational complexity [21]. Therefore, ways in which to balance safety and efficiency deserves more research.

## 6. Conclusions

In this paper, the security of a recent image encryption algorithm called IEA-DESC has been analyzed in detail. It was claimed that some merits of DNA encoding and spatiotemporal chaos are inherited in the algorithm. However, its algorithm structure has several inherent security pitfalls. It was found that IEA-DESC is actually a combined process of DNA-base permutation and bitwise complement from the perspective of cryptanalysis. Therefore, a chosen-plaintext attack and a chosen-ciphertext attack were proposed to recover the equivalent secret key of IEA-DESC, respectively. Both theoretical analysis and experimental results are provided to support effectiveness and efficiency of two attack methods for breaking IEA-DESC. The reported results would help the designers of DNA-based cryptography pay more attention to importance of the essential structure of an encryption scheme, instead of the elegance of the underlying theory.

## Figures and Tables

**Figure 1 entropy-21-00246-f001:**
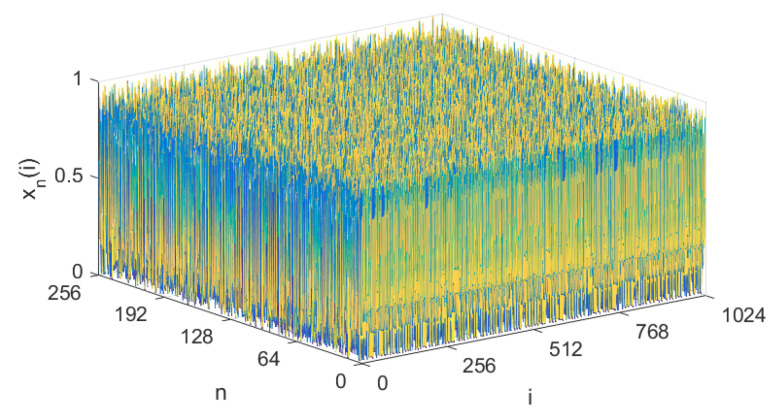
The attractor of the new chaotic algorithm (NCA)-based coupled map lattice (CML).

**Figure 2 entropy-21-00246-f002:**
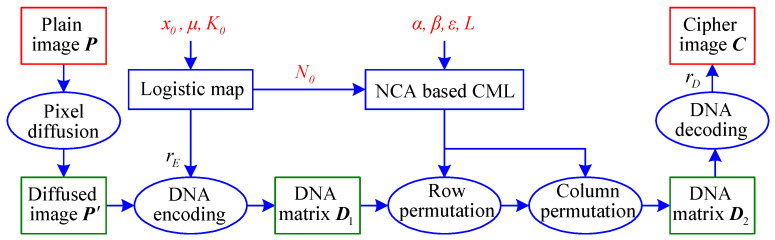
Block diagram of the image encryption algorithm based on DNA encoding and spatiotemporal chaos (IEA-DESC).

**Figure 3 entropy-21-00246-f003:**
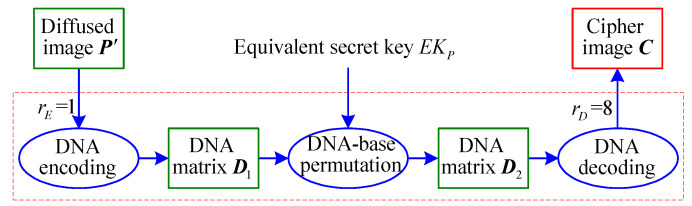
Simplified block diagram of IEA-DESC.

**Figure 4 entropy-21-00246-f004:**
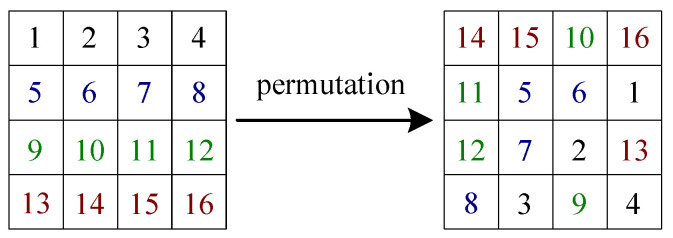
The illustration diagram of a position shuffling for matrices of size 4×4.

**Figure 5 entropy-21-00246-f005:**
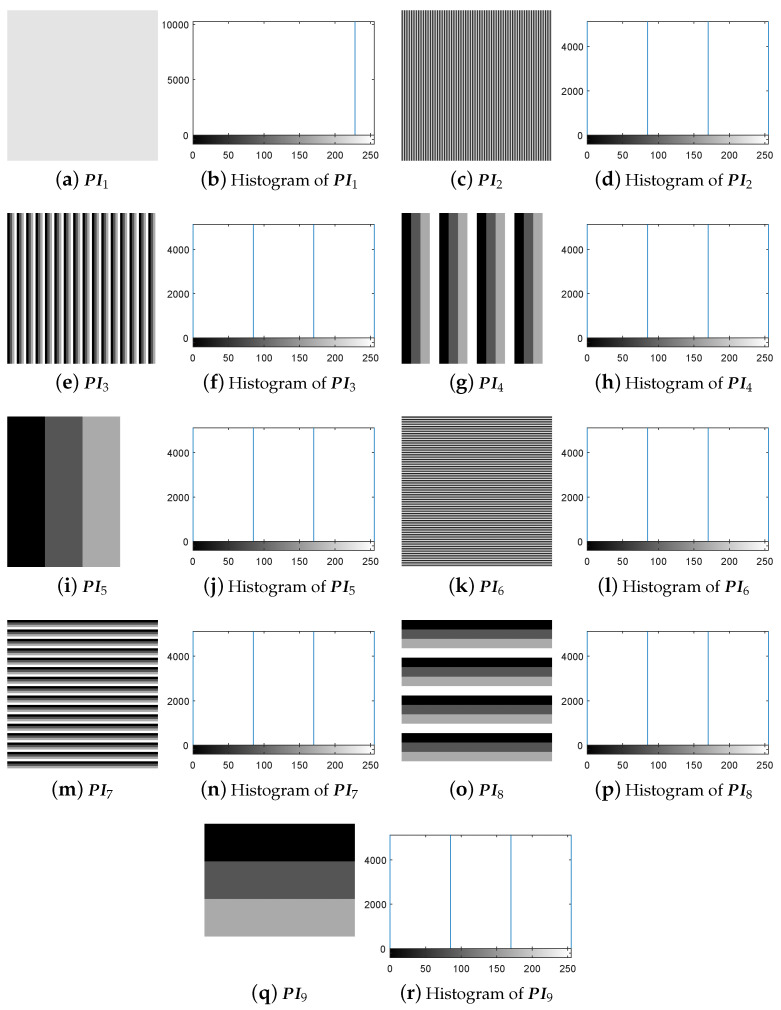
The nine 8-bit special images and their corresponding histograms.

**Figure 6 entropy-21-00246-f006:**
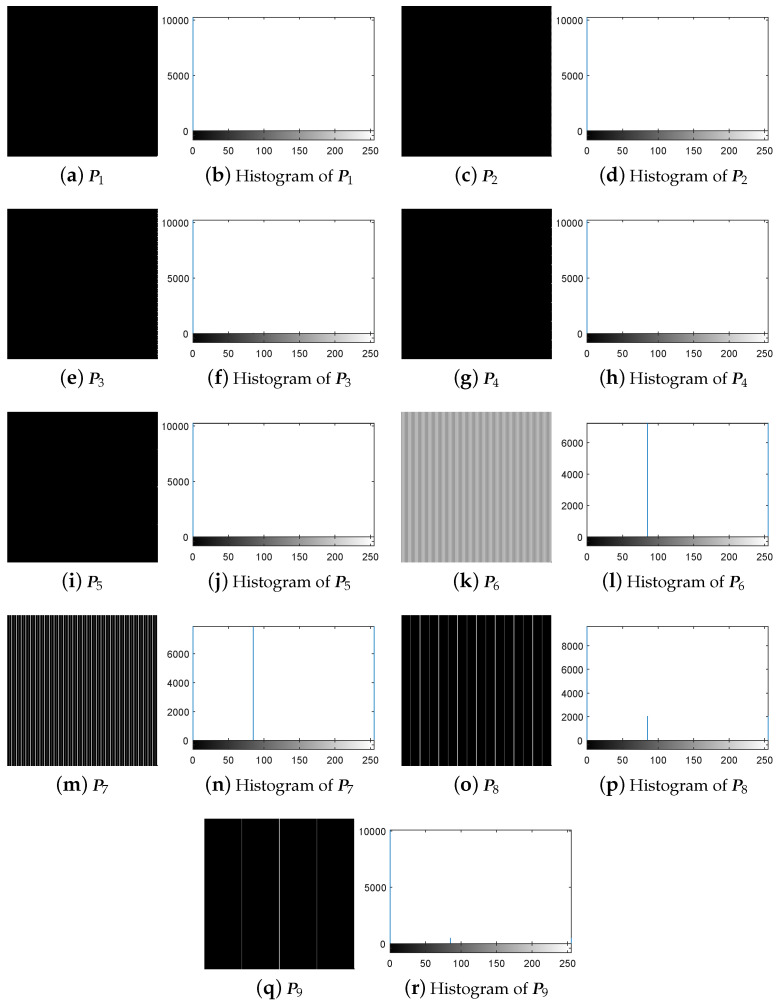
The nine plain-images under chosen-plaintext attack.

**Figure 7 entropy-21-00246-f007:**
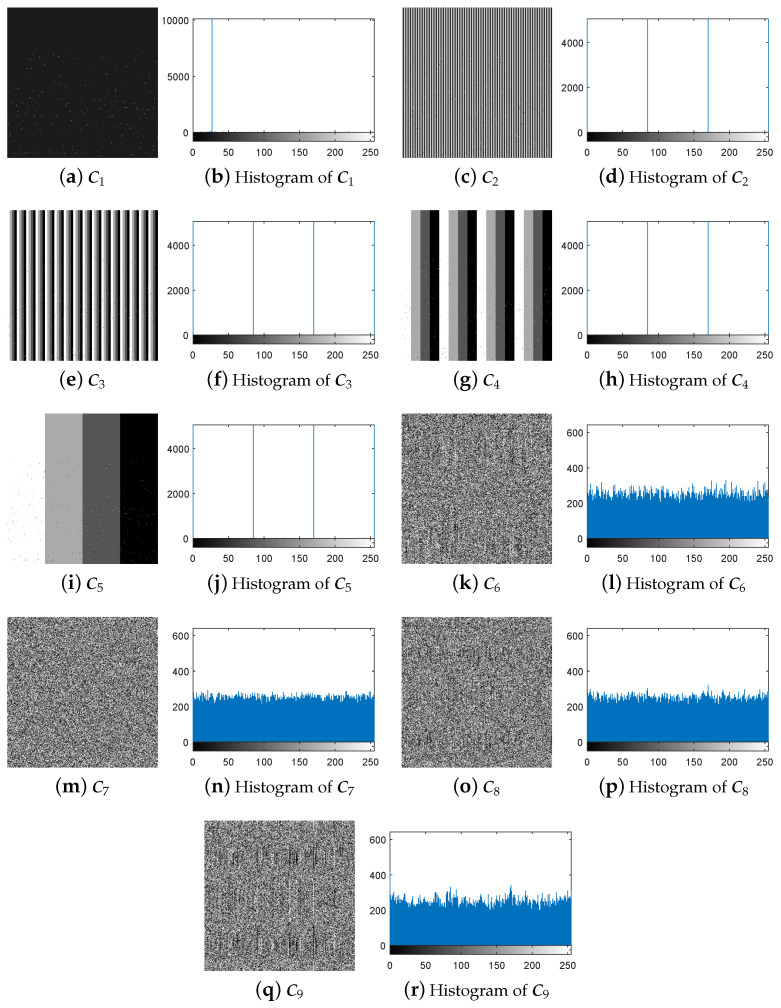
The nine corresponding output cipher-images under chosen-plaintext attack.

**Figure 8 entropy-21-00246-f008:**
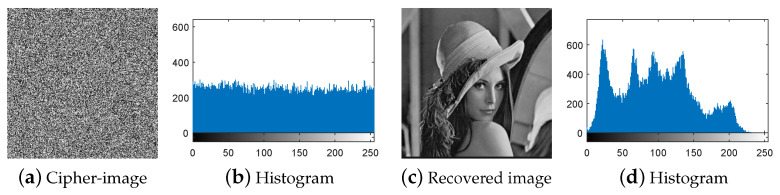
Attacking result on IEA-DESC with the 8-bit image “Lenna”.

**Figure 9 entropy-21-00246-f009:**
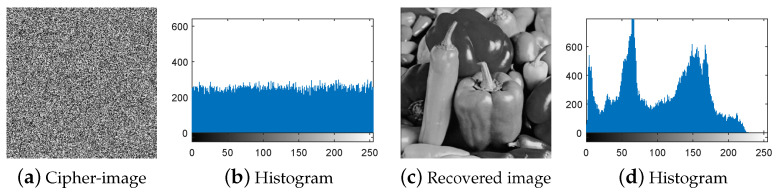
Attacking result on IEA-DESC with the 8-bit image “Peppers”.

**Figure 10 entropy-21-00246-f010:**
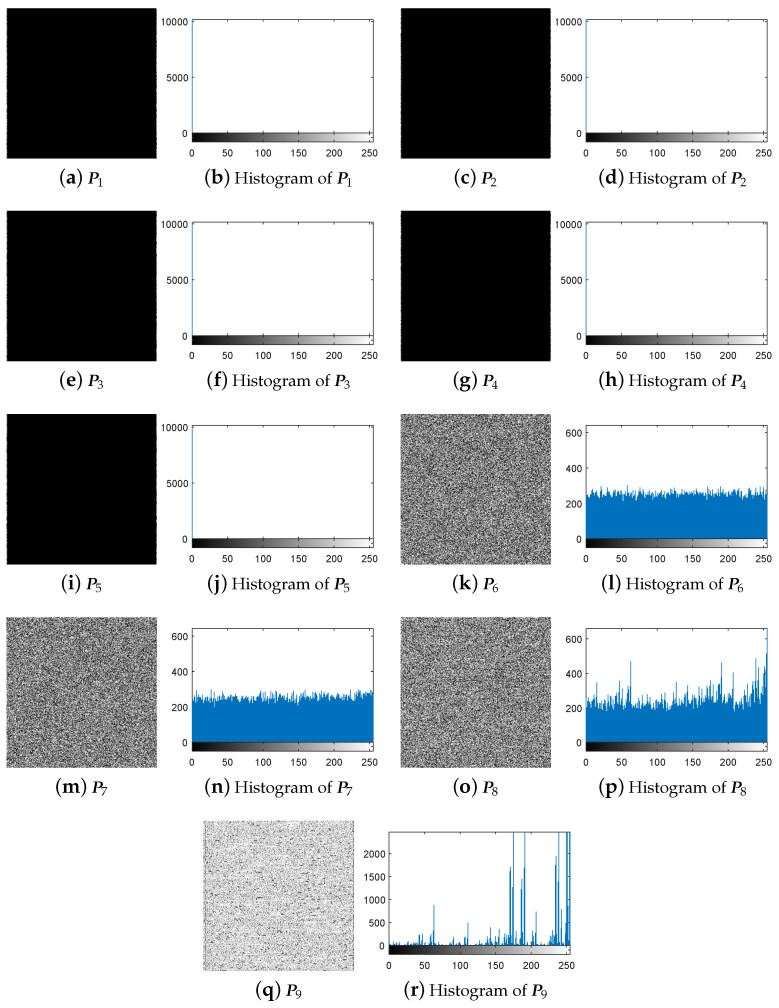
The nine corresponding output plain-images under chosen-ciphertext attack.

**Figure 11 entropy-21-00246-f011:**
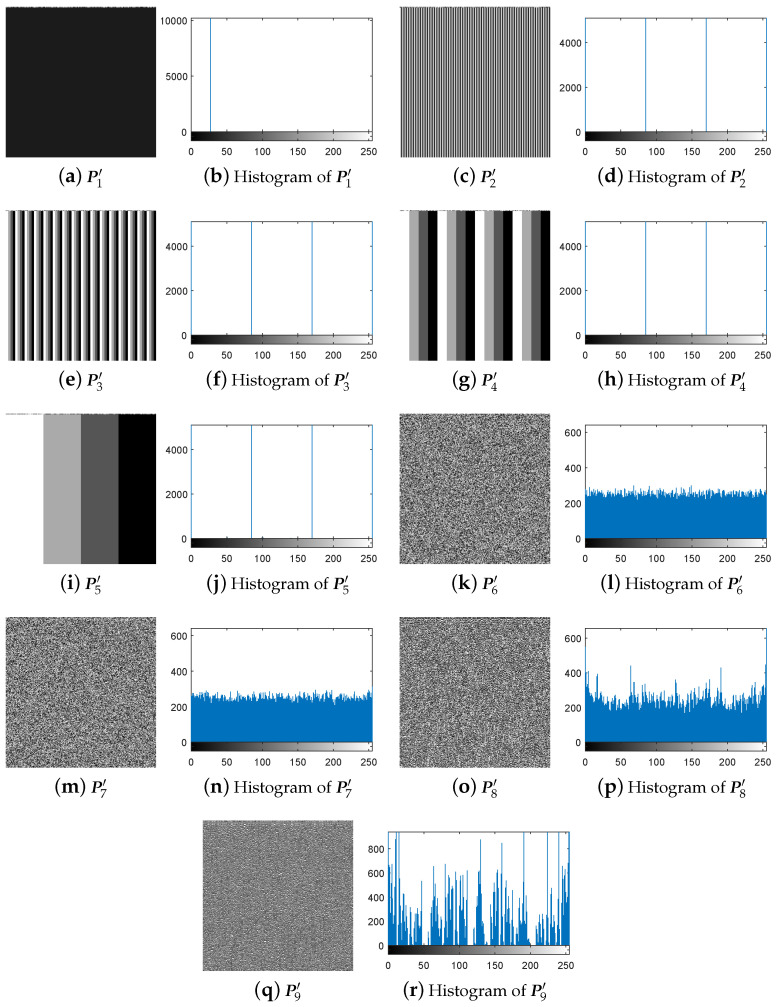
The nine corresponding diffused images under chosen-ciphertext attack.

**Table 1 entropy-21-00246-t001:** Eight kinds of DNA coding rules.

Rules	1	2	3	4	5	6	7	8
A	00	00	01	01	10	10	11	11
T	11	11	10	10	01	01	00	00
G	01	10	00	11	00	11	01	10
C	10	01	11	00	11	00	10	01

**Table 2 entropy-21-00246-t002:** Four common attack methods for cryptanalysis.

Attack Methods	Available Resources for Cryptanalysis
Ciphertext-only attack	The attacker only knows the ciphertext.
Known-plaintext attack	The attacker knows any given plaintext, and also knows the corresponding ciphertext.
Chosen-plaintext attack	The attacker can choose the plaintext that would be useful for deciphering, and also knows the corresponding ciphertext.
Chosen-ciphertext attack	The attacker can choose the ciphertext that is useful for deciphering, and also knows the corresponding plaintext.

**Table 3 entropy-21-00246-t003:** Any 2-bit input and its 2-bit output with the eight pairs of DNA codec rules.

2-Bit Input	DNA-Base with Encoding Rule $r_{E}$								2-Bit Output with Decoding Rule $r_{D}$
	1	2	3	4	5	6	7	8	
00	A	A	G	C	G	C	T	T	11
01	G	C	A	A	T	T	C	G	10
10	C	G	T	T	A	A	G	C	01
11	T	T	C	G	C	G	A	A	00

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
