# Peer review of "Breaking an Image Encryption Algorithm Based on DNA Encoding and Spatiotemporal Chaos"

_entropy, 2019, doi:10.3390/e21030246_

Round 1
Reviewer 1 Report
The paper intends to propose two attack methods based on chosen-plaintext attack and chosen-ciphertext attack are proposed to recover equivalent secret key of IEA-DESC. Both theoretical analysis and experimental results are provided to support effectiveness and efficiency of two attack methods for breaking IEA-DESC.
In my view the paper has many comments as follows:
1- The paper needs several improvements in what concerns the English language. For example, in table 2, third attack method (Chosen-plaintext attack), change “… and also know …” To “… and also knows ….”
2- The authors in introduction didn’t cite any references about image encryption based chaotic economic maps.
3- In equations (9) and (11), change 2^(n-1) to 4^(n-1) according to right hand sides of these equations
Author Response
Response to Reviewer 1 Comments
Dear Reviewer:
Thank you for your letter and for the comments concerning our manuscript entitled "Breaking an Image Encryption Algorithm Based on DNA Encoding and Spatiotemporal Chaos" (ID: entropy-449832). Those comments are all valuable and very helpful for revising and improving our paper, as well as the important guiding significance to our researches. We have studied comments carefully and have made corrections which we hope meet with approval. Revised portion are marked in yellow in the paper. The main corrections in the paper and the responds to the reviewer’s comments are as follows:
Point 1: The paper needs several improvements in what concerns the English language. For example, in table 2, third attack method (Chosen-plaintext attack), change “… and also know …” To “… and also knows ….”
Response 1: According to the reviewer's suggestion, the article was carefully checked and the following corrections were made.
· In table 2, we corrected “… and also know …” to “… and also knows ….”;
· In the last sentences of Sections 4.1 and 4.2, we both corrected “… 8-bit image …” to “… 8-bit images…”;
· Throughout the article, we substituted “key” with “secret key” by finding and replacing method for more precise expression. In addition, we also substituted “spatio-temporal” with “spatiotemporal” for formal unity.
Point 2: The authors in introduction didn’t cite any references about image encryption based chaotic economic maps.
Response 2: Following the comment of the reviewer, we cited two related papers in introduction as follows:
· Entropy 2019, 21, 44. doi:10.3390/e21010044.
· Entropy 2018, 20, 801. doi:10.3390/e20100801.
In addition, we also cited our earlier research below:
· Acta Physica Sinica 2017, 66, 230503. doi:10.7498/aps.66.230503.
Point 3: In equations (9) and (11), change 2^(n-1) to 4^(n-1) according to right hand sides of these equations
Response 3: In equations (9) and (11), we changed “2^(n-1)” to “4^(n-1)” respectively.
Moreover, in order to make the expression clearer, we made a few minor revisions below.
· For more clarity, we updated Figure 1;
· To be clearer, below equations (9) and (11), we corrected “… *Q…” to “… *Q_n …” respectively;
· Below equation (4), we added “and bac rounds the element a to the nearest integer toward minus infinity”.
· Below equation (10), we added “where dae rounds the element a to the nearest integer toward positive infinity”.
· Below equation (2), we changed “and xn(0) = xn(L) is the periodic boundary condition” to “and the periodic boundary condition is xn(0) = xn(L)”.
Last but not least, thanks to you for your good comments.
Reviewer 2 Report
I think that present research is very good organized.
The authors have found some unwanted properties of the iea-desc image encryption algo: low pixel diffusion, not working DNA encoding and decoding, and DNA-based permutation.
Then they investigate the three unwanted properties, first theoretically and then they provide experimental study.
I see that chosen-plaintext attack method and chosen-ciphertext attack methods are about 2.1165s and 2.0785s, respectively. Good time of the proposed breaking methods.
Very interesting attacks. Good job.
Author Response
Response to Reviewer 2 Comments
Dear Reviewer:
Thank you for your letter and for the comments concerning our manuscript entitled "Breaking an Image Encryption Algorithm Based on DNA Encoding and Spatiotemporal Chaos" (ID: entropy-449832). Your encouragement will help us to study harder in the future.
Moreover, after checking carefully again, to be clearer, we made minor revisions as follows:
· In table 2, we corrected “… and also know …” to “… and also knows ….”;
· In the last sentences of Sections 4.1 and 4.2, we both corrected “… 8-bit image …” to “… 8-bit images…”;
· Throughout the article, we substituted “key” with “secret key” by finding and replacing method for more precise expression. In addition, we also substituted “spatio-temporal” with “spatiotemporal” for formal unity;
· In equations (9) and (11), we changed “2^(n-1)” to “4^(n-1)” respectively;
· For more clarity, we updated Figure 1;
· To be clearer, below equations (9) and (11), we corrected “… *Q…” to “… *Q_n …” respectively;
· Below equation (4), we added “and bac rounds the element a to the nearest integer toward minus infinity” ;
· Below equation (10), we added “where dae rounds the element a to the nearest integer toward positive infinity” ;
· Below equation (2), we changed “and xn(0) = xn(L) is the periodic boundary condition” to “and the periodic boundary condition is xn(0) = xn(L)”.
Special thanks to you for your good comments.
